# Oral Health-Related Quality of Life in Patients with Chronic Oral Mucosal Diseases: Reliability and Validity of Urdu Version of Chronic Oral Mucosal Disease Questionnaire (COMDQ)

**DOI:** 10.3390/healthcare11040606

**Published:** 2023-02-17

**Authors:** Danial Qasim Butt, Osama Khattak, Farooq Ahmad Chaudhary, Alzarea K. Bader, Hasan Mujtaba, Azhar Iqbal, Shahab Ud Din, Gharam Radhi A. Alanazi, Mohammed Ghazi Sghaireen, Waseem Ahmad

**Affiliations:** 1School of Dental Sciences, Universiti Sains Malaysia, Kubang Kerian 16150, Malaysia; 2Department of Restorative Dentistry, Jouf University, Sakaka 72388, Saudi Arabia; 3School of Dentistry (SOD), Shaheed Zulfiqar Ali Bhutto Medical University (SZABMU), Islamabad 44000, Pakistan; 4Department of Prosthetic Dental Sciences, College of Dentistry, Jouf University, Sakaka 72388, Saudi Arabia; 5College of Dentistry, Jouf University, Sakaka 72388, Saudi Arabia

**Keywords:** chronic oral mucosal diseases, COMDQ, oral health-related quality of life, quality of life, reliability, validity

## Abstract

The aim of the study was to translate and validate the Urdu version of the Chronic Oral Mucosal Disease Questionnaire (COMDQ) and measure the oral health-related quality of life (OHRQoL) in Pakistani patients with chronic oral mucosal disease. One hundred and twenty patients with chronic oral mucosal diseases were recruited for this study. Two types of reliability of the COMDQ were tested. First, the internal consistency was calculated using Cronbach’s alpha, and second, test-retest reliability was calculated using intraclass correlation coefficients (ICC). Convergent validity was assessed for testing the validity of the COMDQ by examining the correlations with the Visual Analogue Scale (VAS) and OHIP-14 using Pearson’s correlations coefficient, and a t-test was used to compare the COMDQ domains and the socio-demographic characteristics. The most prevalent chronic oral mucosal disease (COMD) among the participants was recurrent aphthous stomatitis (47.5%), and the least prevalent was oral granulomatosis (6.6%). The total mean score for COMDQ was 43.5 (SD = 18.4). It showed a high level of internal consistency (Cronbach’s a = 0.81), and test-retest reliability was also good (r = 0.85). The total score of COMDQ was strongly correlated with the total score of OHIP-14 and VAS (r = 0.86 and r = 0.83), which indicated good convergent validity. The score of pain and the functional limitation domain reported a significant difference with age (*p* < 0.021) and employment status (*p* < 0.034). The Urdu version of COMDQ is an accurate, valid, and reliable instrument that can be used to assess the OHRQoL in patients with chronic oral mucosal diseases in Pakistani and other Urdu-speaking populations of different age groups.

## 1. Introduction

Chronic oral mucosal disease refers to a wide range of autoimmune, infectious, and inflammatory states of the soft tissue of the oral cavity. These diseases include recurrent aphthous stomatitis, oral lichen planus, mucous membrane pemphigoid, pemphigus vulgaris, and oral granulomatosis [1]. These diseases are rarely life-threatening; however, they cause significant disruption and discomfort in patients’ physical, social, and psychological states due to their painful, long-standing, and recurrent nature, which can impact their overall quality of life (QoL) [2,3]. Furthermore, the use of topical or systemic medications (corticosteroids and immunosuppressant agents) for the treatment of these chronic diseases with their side effects can negatively affect patients’ QoL and OHRQoL [4,5]. Therefore, it is recommended that the assessment of how the disease and its management affect QoL should be included in the clinical assessment and therapeutic interventions [6]. It can facilitate a patient’s experience with specific interventions and enable easier communication with patients, as the ultimate clinician’s therapeutic goal is to enhance the patient’s QoL.

Tools used for the assessment of QoL can be divided into generic and disease-specific tools [7]. The main benefit of using generic QoL measures is that they allow the comparison of the results with several different diseases; however, it is generally unable to detect the minor clinical changes associated with a specific disease [6,8]. Disease-specific instruments are increasingly acknowledged and used by researchers as they are more sensitive and can accurately detect small and specific disease-related changes. The Chronic Oral Mucosal Disease Questionnaire (COMDQ) is a disease-specific instrument created by Ni Riordin in Ireland in 2011 for use in oral medicine and radiology departments to measure the QoL in chronic mucosal disease [3,9]. It is recommended that to use the questionnaire in a different language with different social and cultural structures, the questionnaire should be translated and validated to preserve its integrity and obtain reliable data [10]. This COMDQ has been translated and validated in various languages; however, it is not available in the Urdu language and has never been used in the Pakistani population. Hence, this study aimed to translate and validate the Urdu version of COMDQ and measure the QoL in Pakistani patients with chronic oral mucosal disease.

## 2. Materials and Methods

The patients with chronic mucosal disease were conveniently recruited for this cross-sectional study from the Oral Medicine Department of the School and Hospital of Dentistry, Shaheed Zulfiqar Ali Bhutto Medical University, between January and May 2022. Patients older than 18 years, diagnosed with chronic oral mucosal disease based on history, examination, hematological and histological tests, and who underwent any treatment for COMDs were invited to participate in this study. The recruits were briefed about the study, and written informed consent was taken from them. The required approval was taken from the ethical review committee of SZABMU (Ref. no. SOD/ERB/2022/264). The recommended sample size mentioned in the literature for studies related to the validation of questionnaires is between 100–200 [11]. Therefore, in this study, 120 chronic oral mucosal disease patients were recruited.

The COMDQ questionnaire consists of 26 items that are divided into four domains, Pain and Functional limitation (9 items), Medication and Treatment (6 items), Social and Emotional (7 items), and Patient Support (4 items). The responses were scored on a 5-point Likert scale from “not at all” (0) to “extremely” (4). The calculation of the final score of COMDQ is carried out by adding the score of all the items ranging from 0 to 104, where a lower score indicates poor quality of life regarding COMDs [12,13].

### 2.1. Translation Process

Published steps and guidelines were followed for the translation and cross-cultural adaptation of the COMDQ [14,15,16]. The steps include forward translation, committee review, back-translation, and second committee review. In the first step, two bilingual translators translated the English version of COMDQ into the Urdu language. The forward-translated version was then examined by a four-member expert committee consisting of specialists in psychology, translation, statistics, and oral medicine. After a comprehensive review by experts, the forward-translated Urdu version was back-translated by two separate translators. That version was again reviewed by the committee that compared the back-translated version with the original one and checked its clarity, cultural acceptability, and conceptual equivalence. In the final step, the final committee-approved Urdu version of the COMDQ was pilot tested in 20 patients to verify its clarity and acceptance. The results showed that all the participants clearly understood the questions; therefore, no further changes were made, and the final version of Urdu COMDQ was finalised.

### 2.2. Questionnaires and Tools

The Oral Health Impact Profile-14 consists of 14 items that measure the difficulties experienced by patients related to oral health in the past in seven domains. The domains are functional limitations, physical pain, psychological discomfort, physical disability, psychological disability, social disability, and handicaps. The participant responded to each item on a 5-point Likert scale of 0 (never), 1 (seldom), 2 (sometimes), 3 (often), or 4 (very often). The total score is calculated by adding the score of 14 items, which range from 0 to 56, with a higher score indicating a worse quality of life [17,18].

The Visual Analogue Scale (VAS) is a simple analogue scale containing a 10-cm (100 mm) line with ‘no pain’ marked at 0 cm and ‘worst imaginable pain’ marked at 10 cm. The participant marked the line to indicate the severity of their pain, with a higher score indicating greater pain [19,20]. 

The analysis was carried out using SPSS version 25.0. Socio-demographic and clinical characteristics were presented using means, frequencies, and percentages. Two types of reliability of the COMDQ were tested. First, the internal consistency was calculated using Cronbach’s alpha, and second, test-retest reliability was calculated using intraclass correlation coefficients (ICC). For testing the validity of the COMDQ, convergent and construct validity were assessed. It was expected that participants with higher VAS scores would have a poor score on the COMDQ, and the COMDQ score of the participants would correlate with the OHIP-14 score. These relationships were assessed using Pearson’s correlation coefficient. The t-test was used to compare the COMDQ domains and the socio-demographic characteristics. A value of *p* < 0.05 was set as significant for data analysis.

## 3. Results

A total of 120 patients were recruited for this study. The mean age of the participants was 43.2 (SD = 12.4), and the majority were above the age of 35 years (67.5%), female (62.5%), and employed (69.1%). The most prevalent COMD among the participants was recurrent aphthous stomatitis (47.5%), and the least prevalent was oral granulomatosis (6.6%). The total mean score for the Urdu version of COMDQ was 43.5 (SD = 18.4), and for the domain score of Pain and Function, the highest mean score of 13.2 (SD = 8.7) was reported. For the OHIP-14, the total mean score was 14.5 (SD = 11.3), with the most severely impacted domain being physical pain (mean = 3.23, SD = 2.5) (Table 1).

Table 2 presents the results of the reliability analysis of COMDQ. The COMDQ showed good reliability, with the Cronbach’s alpha value of its domains ranging from 0.79 to 0.92. The Cronbach’s alpha value for the total score was 0.81. The retest was carried out on 40 patients after a two-week interval, and the test-retest reliability results also showed good reliability, with an ICC value of COMDQ domains ranging between 0.78 to 093. The ICC value for the total score was 0.85.

The Pearson correlation results are shown in Table 3. The results showed positive and strong significance between the different domains of the COMDQ, OHIP, and VAS. The range of correlation coefficients was between 0.45 to 0.86, which indicated good convergent validity. The total score of COMDQ was strongly correlated with the total score of OHIP-14 and VAS (r = 0.86 and r = 0.83).

The scores of the pain and functional limitation domains reported a significant difference with age (*p* < 0.021) and employment status (*p* < 0.034). The mean score was significantly higher in patients with ages above 35 years (mean = 15.5) and unemployed (mean = 14.3). The score for the social and emotional domains reported a significant difference with all the socio-demographic characteristics (*p* < 0.05) (Table 4).

## 4. Discussion

Recently, quality-of-life instruments have frequently been used as outcomes that measure the impact of oral health conditions in various patients having diverse cultures and languages. Hence, the demand has increased for the cross-cultural adaptation and validation of these measures. Pakistan ranks fifth in the world by population, and its language, Urdu, is the 11th most widely spoken language not only in Pakistan but also in other south Asian countries [21]. This necessitates and emphasises the development of the Urdu version of the COMDQ. Therefore, in this study, for the first time, an Urdu version of the COMDQ was developed to assess OHRQoL, and its psychometric properties were tested in Pakistani patients with chronic oral mucosal diseases.

The instrument became an appropriate tool in terms of psychometrics when the standard published guidelines were used for the process of translation and validation. In this study, the translation of COMDQ was carried out using the common technique of forward translation, committee review, backward translation, and committee review, so the items were adapted without any problems [14,15,22].

In this study, the COMDQ demonstrated good internal consistency (Cronbach’s α = 0.81), which is approximately similar to the Irish and Chinese (Cronbach’s α = 0.89) versions of COMDQ [1,23,24,25]. In this study, along with the Irish and Chinese studies, the patient support domain showed lower internal consistency with other domains, which might indicate that patients with chronic oral mucosa disease did not receive sufficient support and were neglected. For the reliability of the test-retest of COMDQ, the intraclass correlation coefficients exhibit a very good standard of 0.85, indicating a high level of stability, compatibility, and reproducibility at different time effects. This ICC result was consistent with the above-mentioned Chinese (ICC = 0.83) and Irish (ICC = 0.86) studies [1,23]. 

In this study, the convergent validity of the COMDQ score with the OHIP-14 and VAS was good, with Pearson correlation coefficients of 0.86 with OHIP-14 and 0.83 with the VAS score. These values reflected very good validity, and these values are even higher than those reported in an Irish study [12]. 

In this study, the patients under 35 years of age scored significantly lower than the patients over 35 years of age in the domain of pain and functional limitations. However, the study carried out by Bijina et al. on the QoL in patients with COMD by COMDQ reported opposite results, with older patients scoring significantly lower in the pain and functional limitations domain, and similar results observed in the study on the Turkish population [6,26]. A small proportion of participants with ages lower than 35 years in this study would be a plausible reason for this difference in the results. Furthermore, in this study, the female patients showed significantly higher scores than males in the social and emotional domains. This result is consistent with the results of the Turkish study [26]. This study also demonstrated that in unemployed patients, the score was significantly higher than the employed ones in both pain and functional and social and emotional domains. Similar results were seen in the Turkish population, with the exception of the medication and treatment domain, which showed a significant difference in scores between employed and unemployed in the Turkish population; however, it was non-significant in this study [26].

There are some limitations associated with this study. Firstly, the sample has very few patients with mucous member pemphigoid, pemphigus vulgaris, and oral granulomatosis. This issue can be resolved in future studies by taking a larger sample size. The analysis of the dimensional structure of the scale was not performed due to time constraints. Secondly, the many socioeconomic factors, such as education level and income status, that can influence QoL with patients’ perception were not assessed. Lastly, the sample was recruited from one oral medicine department of a medical iniversity, which may not represent the whole COMD population. Future studies can use a longitudinal study design to see the effect of medications on the QoL of COMD patients.

## 5. Conclusions

The findings of this study demonstrated that the Urdu version of COMDQ is an accurate, valid, and reliable instrument that can be used to assess the QoL in patients with chronic oral mucosal diseases in Pakistani and other Urdu-speaking populations of different age groups. Furthermore, chronic oral mucosal diseases are common in Pakistan, and they can impact patients’ quality of life and well-being. 

## Figures and Tables

**Table 1 healthcare-11-00606-t001:** Characteristics of patients and descriptive statistics of COMDQ, OHIP-14, and VAS.

Characteristics	N (%)
Age <35 years old >35 years old	Mean (SD) 43.2 (12.4) 39 (32.2) 81(67.5)
Gender Male Female	45 (37.5) 75 (62.5)
Employment Employed Unemployed	83 (69.1) 37 (30.9)
Clinical classification Recurrent aphthous stomatitis Oral lichen planus Mucous membrane pemphigoid Pemphigus vulgaris Oral granulomatosis	57 (47.5) 35 (29.1) 11 (9.1) 9 (7.5) 8 (6.6)
Questionnaires	Mean (SD)
COMDQ Pain and function limitation Medication and treatment Social and emotional Patient support Total score	13.2 (8.7) 10.5 (6. 3) 12.6 (8.2) 7.2 (4.5) 43.5 (18.4)
OHIP-14 Functional limitations Physical pain Psychological discomfort Physical disability Psychological disability Social disability Handicap Total	1.51 (1.7) 3.23 (2.5) 2.14 (2.2) 2.57 (2.1) 2.24 (2.2) 1.65 (1.8) 1.23 (1.3) 14.5 (11.3)
VAS Pain	4.47 (2.78)

**Table 2 healthcare-11-00606-t002:** Reliability analysis of the Urdu version of COMDQ.

Domain	Cronbach’s Alpha	Intraclass Correlation Coefficient
Pain and function limitation	0.82	0.78
Medication and treatment	0.79	0.85
Social and emotional	0.88	0.93
Patient support	0.92	0.81
Total score	0.81	0.85

**Table 3 healthcare-11-00606-t003:** Correlation between COMDQ, OHIP-14 and VAS.

COMDQ Domains	OHIP-14 Domains							Analogue Scale	OHIP Total Score
	Functional Limitations	Physical Pain	Psychological Discomfort	Physical Disability	Psychological Disability	Social Disability	Handicap Visual		
Pain and function limitation	0.61	0.68	0.51	0.61	0.49	0.52	0.53		
Medication and treatment	0.64	0.57	0.48	0.56	0.53	0.61	0.56		
Social and emotional	0.53	0.59	0.68	0.64	0.74	0.70	0.54		
Patient support	0.45	0.46	0.45	0.46	0.49	0.47	0.46		
Visual Analogue Scale	0.67	0.72	0.58	0.63	0.60	0.63	0.64		
OHIP total score	0.70	0.79	0.77	0.78	0.74	0.75	0.72	0.74	
COMDQ total score	0.65	0.69	0.61	0.73	0.68	0.68	0.64	0.83	0.86

**Table 4 healthcare-11-00606-t004:** The summary of the relationship of socio-demographic characteristics with domains of COMDQ.

Domain	Age < 35	Age > 35	Male	Female	Employed	Unemployed
Pain and function limitation	11.2 (8.5)	15.5 (9.1)	12.6 (6.1)	13.9 (7.4)	10.2 (4.6)	14.3 (5.7)
*p*-value	0.021 *		0.52		0.034 *	
Medication and treatment	10.6 (5.4)	11.2 (7.9)	9.1 (6.9)	10.3 (7.9)	8.76 (5.3)	9.6 (4.8)
*p*-value	0.084		0.17		0.093	
Social and emotional	11.5 (7.7)	15.8 (8.3)	10.8 (9.2)	14.6 (6.5)	11.1 (6.3)	14.2 (7.4)
*p*-value	0.041 *		0.03 *		0.044 *	
Patient support	5.51 (2.3)	6.7.3 (3.5)	6.1 (3.2)	6.8 (4.3)	6.2 (3.8)	6.7 (4.3)
*p*-value	0.078		0.84		0.28	

* Statistically significant (*p* < 0.05).

## Data Availability

The data presented in this study are available on request from the corresponding author.

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
