# Peer review of "Oral Health-Related Quality of Life in Patients with Chronic Oral Mucosal Diseases: Reliability and Validity of Urdu Version of Chronic Oral Mucosal Disease Questionnaire (COMDQ)"

_healthcare, 2023, doi:10.3390/healthcare11040606_

Round 1

Reviewer 1 Report

Good morning. The submitted paper aims to evaluate and validate the URDU language translation of the COMDQ.

The paper does not have a high scientific impact and may be of limited interest, with the understanding that of course it is always best to administer medical questionnaires in one's native language. Having said that, I think that in the Materials and Methods section it should be better explained what was intended to be done: on the one hand the translation of the questionnaire into the URDU language with all the translation and verification steps, and on the other hand its validation by comparing it with other questionnaires already in use. This step is not well explained and should be improved. Also the questionnaires used, especially the OHIP-14 , should be better presented and explained. The paper, also, although written quite well, needs, in my opinion, a careful re-reading for grammatical and spelling improvement (punctuation, too long sentences, etc.). Also, it is always mandatory, the first time, not to use acronyms but the full definition (e.g. line 24, pag 1)

Author Response

Comment: I think that in the Materials and Methods section it should be better explained what was intended to be done: on the one hand the translation of the questionnaire into the URDU language with all the translation and verification steps, and on the other hand its validation by comparing it with other questionnaires already in use. This step is not well explained and should be improved.

Also the questionnaires used, especially the OHIP-14 , should be better presented and explained. The paper, also, although written quite well, needs, in my opinion, a careful re-reading for grammatical and spelling improvement (punctuation, too long sentences, etc.). Also, it is always mandatory, the first time, not to use acronyms but the full definition (e.g. line 24, pag 1)

Response: Material and methods sections is revised and improved for better understanding as suggested.

The presentation of OHIP-14 questionnaire is revised and improved as suggested.

The manuscript is thoroughly re-checked for grammatical and spelling mistakes and the full definition of acronyms were written and corrected.

Reviewer 2 Report

Congrats for the wonderful paper!! There are some details that could benefit the article. Information is shown duplicate in tables, if appear in text, delete it. Clonclusion must be clear and concise, it must answer the main objective , no more 2-3 lines.

Author Response

Comment: Congrats for the wonderful paper!! There are some details that could benefit the article. Information is shown duplicate in tables, if appear in text, delete it. Conclusion must be clear and concise, it must answer the main objective , no more 2-3 lines.

Response: Thank you and noted. The results and conclusion section is revised and improved as suggested.

Reviewer 3 Report

The purpose of this study was to translate and validate the Urdu version of the COMDQ, and to measure oral health-related quality of life in Pakistani patients with chronic oral mucosal disease. This article is timely and well written. The validation method is within the framework of classical test theory and is fully justified. The authors are aware of their small sample size, which limits the significance of their results. It is unfortunate that an analysis of the dimensional structure of the scale was not performed in order to compare it to the original scale.  This point should be discussed. 

Author Response

Comment: The purpose of this study was to translate and validate the Urdu version of the COMDQ, and to measure oral health-related quality of life in Pakistani patients with chronic oral mucosal disease. This article is timely and well written. The validation method is within the framework of classical test theory and is fully justified. The authors are aware of their small sample size, which limits the significance of their results. It is unfortunate that an analysis of the dimensional structure of the scale was not performed in order to compare it to the original scale.  This point should be discussed.

Response: Thank you and noted. Mentioned point is added in the discussion section as suggested.

Reviewer 4 Report

Dear author

The purpose of this study was to translate, validate the Urdu version of COMDQ, and measure the oral health-related quality of life (OHRQoL) in Pakistani patients with the chronic oral mucosal disease. One hundred and twenty patients with chronic oral mucosal diseases were recruited for this study.

I don't see any major issues with this article. Please add the following two points.

1.Please explain the rationale for the sample size.

2.P6L178-190 Please consider why the results differed from previous studies.

Author Response

Comment: Dear author, The purpose of this study was to translate, validate the Urdu version of COMDQ, and measure the oral health-related quality of life (OHRQoL) in Pakistani patients with the chronic oral mucosal disease. One hundred and twenty patients with chronic oral mucosal diseases were recruited for this study.

I don't see any major issues with this article. Please add the following two points.

1.Please explain the rationale for the sample size.

2.P6L178-190 Please consider why the results differed from previous studies.

Response: Thank you. The rationale of sample size is mentioned in methods section (line 77-79). The difference in the results is discussed and added in the discussion section as suggested.